# Cabbage Sweeter than Cake? Analysing the Potential of Large Language Models for Learning Conceptual Spaces

**Usashi Chatterjee, Amit Gajbhiye, Steven Schockaert**
CardiffNLP, Cardiff University, UK
{chatterjeeu,gajbhiyea,schockaerts1}@cardiff.ac.uk

## Abstract

The theory of Conceptual Spaces is an influential cognitive-linguistic framework for representing the meaning of concepts. Conceptual spaces are constructed from a set of quality dimensions, which essentially correspond to primitive perceptual features (e.g. hue or size). These quality dimensions are usually learned from human judgements, which means that applications of conceptual spaces tend to be limited to narrow domains (e.g. modelling colour or taste). Encouraged by recent findings about the ability of Large Language Models (LLMs) to learn perceptually grounded representations, we explore the potential of such models for learning conceptual spaces. Our experiments show that LLMs can indeed be used for learning meaningful representations to some extent. However, we also find that fine-tuned models of the BERT family are able to match or even outperform the largest GPT-3 model, despite being 2 to 3 orders of magnitude smaller.[1]

## 1 Introduction

Conceptual spaces (Gärdenfors, 2000) represent concepts in terms of cognitively meaningful features, called quality dimensions. For example, a conceptual space of colour is composed of three quality dimensions, representing hue, saturation and intensity. Conceptual spaces provide an elegant framework for explaining various cognitive and linguistic phenomena (Gärdenfors, 2014). Within Artificial Intelligence (AI), the role of conceptual spaces is essentially to act as an intermediate representation layer, in between neural and symbolic representations (Gärdenfors, 2004). As such, conceptual spaces could play a central role in the development of explainable AI systems. Unfortunately, such representations are difficult to learn from data.

Most applications of conceptual spaces are thus limited to narrow domains, where meaningful representations can be learned from ratings provided by human participants (Paradis, 2015; Zwarts, 2015; Chella, 2015).

In this paper, we explore whether Large Language Models (LLMs) could be used for learning conceptual spaces. This research question is closely related to the ongoing debate about the extent to which Language Models (LMs) can learn perceptually grounded representations (Bender and Koller, 2020; Abdou et al., 2021; Patel and Pavlick, 2022; Søgaard, 2023). Recent work seems to suggest this might indeed be possible, at least for the colour domain. For instance, Abdou et al. (2021) found that LMs are able to learn representations of colour terms which are isomorphic to perceptual colour spaces. When it comes to predicting the typical colour of objectes, Paik et al. (2021) found that the predictions of LMs are heavily skewed by surface co-occurrence statistics, which are unreliable for colours due to reporting bias (Gordon and Durme, 2013), i.e. the fact that obvious colours are rarely mentioned in text. However, Liu et al. (2022a) found the effects of reporting bias to largely disappear in recent LLMs. These findings suggest that it may now be possible to distill meaningful conceptual space representations from LLMs, as long as sufficiently large models are used. However, existing analyses are limited in two ways:

- Several works have explored the colour domain, and visual domains more generally (Li et al., 2023), but little is known about the abilities of LLMs in other perceptual domains.

- Most analyses focus on classifying concepts, e.g. predicting colour terms or the materials from which objects are made, rather than on evaluating the underlying quality dimensions.

We address the first limitation by including an evaluation in the taste domain. To address the second

---

[1] Our datasets and evaluation scripts are available at `https://github.com/ExperimentsLLM/EMNLP2023_PotentialOfLLM_LearningConceptualSpace`.

limitation, rather than considering discrete labels (e.g. *sweet*), we use LLMs to rank concepts according to the degree to which they have a particular feature (e.g. *sweetness*).

## 2 Datasets

The primary focus of our experiments is on the taste domain, which has not yet been considered in this context, despite having a number of important advantages. For instance, the relevant quality dimensions are well-established and have a linear structure (unlike hue in the colour domain). This domain also seems particularly challenging, as the typical terms which are used to describe taste only apply to extreme cases. For instance, we can assert that grapefruit is bitter and bananas are sweet, but it is less clear how a language model would learn whether chicken is sweeter than cheese. As ground truth, we rely on the ratings that were collected by Martin et al. (2014). They rated a total of 590 food items along six dimensions: sweet, salty, sour, bitter, umami and fat. The ratings were obtained by a panel of twelve assessors who were experienced in sensory profiling. They scored the food items during an eight month measurement phase, after having received 55 hours of training in a laboratory. We manually rephrased some of the names of the items in this dataset, to make them more natural. For instance, *cherry (fresh fruit)* was changed to *cherry* and *hake (grilled with lemon juice)* was changed to *grilled hake with lemon juice*.

We complement our analysis in the taste domain with experiments on three basic physical domains: mass, size and height. These were found to be particularly challenging by Li et al. (2023), with LLMs often failing to outperform random guessing. As the ground truth for mass, we use the household dataset from Standley et al. (2017), which specifies the mass of 56 household objects. The original dataset includes images of each object. We removed 7 items which were not meaningful without the image, namely *big elephant*, *small elephant*, *Ivan's phone*, *Ollie the monkey*, *Marshy the elephant*, *boy doll* and *Dali Clock*, resulting in a dataset of 49 objects. We treat this problem as a ranking problem. Li et al. (2023) also created a binary classification version of this dataset, which involves judging pairwise comparisons (e.g. is a red lego brick heavier than a hammer?). For size and height, we use the datasets created by Liu et al. (2022b). These size and height datasets each con-

sist of 500 pairwise judgements (e.g. an ant is larger than a bird). Note that unlike for the other datasets, no complete ranking is provided.

## 3 Methods

We experiment with a number of different models.

**Ranking with GPT-3**  We use GPT-3 models of four different sizes[2]: *ada*, *babbage*, *curie* and *davinci*. To rank items according to a given dimension, we use a prompt that contains the name of that dimension as the final word, e.g. for sweetness we could use *"It is known that [food item] tastes sweet"*. We then use the probability of this final word, conditioned on the rest of the prompt, to rank the item: the higher the probability of *sweet*, the more we assume the item to be sweet.

**Pairwise Comparisons with GPT-3**  To predict pairwise judgements, we consider two approaches. First, we again use conditional probabilities. For instance, to predict whether *an ant is larger than a bird*, we would get the conditional probability of *large* in the sentences *an ant is large* and *a bird is large*. If the conditional probability we get from the first sentence is lower than the probability from the second sentence, we would predict that the claim that *an ant is larger than a bird* is false. Second, we use a prompt that asserts the statement to be true (e.g. "An ant is larger than a bird") and a prompt that asserts the opposite (e.g. "A bird is larger than an ant"). We compute the perplexity of both statements and predict the version with the lowest perplexity to be the correct one.

**Ranking with ChatGPT and GPT-4**  ChatGPT and GPT-4 are more difficult to use than GPT-3 because the OpenAI API does not allow us to compute conditional probabilities for these models. Instead, to use these conversational models, we directly ask them to rank a set of items, using a prompt such as: *Rank the following items according to their size, from the largest to the smallest*, followed by a list of items to be ranked.

**Baseline: DeBERTa**  We consider two baselines. First, we use a DeBERTa-v3-large model (He et al., 2021), which we fine-tuned to predict the commonsense properties of concepts. To this end, we

---

[2]The exact model sizes have not been made public, but were estimated to be 350M parameters for *ada*, 1.3B parameters for *babbage*, 6.7B parameters for *curie* and 175B parameters for *davinci*: https://blog.eleuther.ai/gpt3-model-sizes/.

used the extended McRae dataset (McRae et al., 2005) introduced by Forbes et al. (2019) and the augmented version of CSLB[3] introduced by Misra et al. (2022). Together, these two datasets contain 19,410 positive and 31,901 negative examples of (concept,property) pairs. We fine-tune the model on these examples using the following prompt: *can* `concept` *be described as* `property`? *<MASK>*. For instance, for the example (banana,yellow), the corresponding prompt would be: *can banana be described as yellow? <MASK>*. The probability that the concept has the property is then predicted using a linear classifier that takes the final-layer embedding of the <MASK> token as input. We use the resulting model for the different evaluations, without any further fine-tuning.

**Baseline: Bi-encoder**  As the second baseline, we use two variants of the bi-encoder model from Gajbhiye et al. (2022). First, we use the original BERT-large model from Gajbhiye et al. (2022) that was trained on data from Microsoft Concept Graph (Ji et al., 2019) and GenericsKB (Bhakthavatsalam et al., 2020). However, as these training sets are not specifically focused on commonsense knowledge, we used ChatGPT to construct a dataset of 109K (concept,property) pairs, since no existing dataset of sufficient size and quantity was available. The key to obtain high-quality examples was to ask the model to suggest properties that are shared by several concepts, and to vary the examples that were provided as part of a few-shot prompting strategy. More details on how we collected this dataset using ChatGPT are provided in Appendix A. We then trained the BERT-large bi-encoder on this dataset.

## 4 Experiments

**Taste Domain**  Table 1 summarises the main results on the taste dataset. For this experiment, each model was used to produce a ranking of the 590 food items, which was then compared with the ground truth in terms of Spearman's rank correlation ($\rho$). For the LMs, we experimented with two prompts. Prompt 1 is of the form "[food item] tastes [sweet]", e.g. "apple tastes sweet". Prompt 2 is of the form "it is known that [food item] tastes [sweet]". For the DeBERTa model and the bi-encoders, we verbalised the sweetness property as "tastes sweet", and similar for the others. The results in Table 1 show a strong correlation, for

| | | Sweet | Salty | Sour | Bitter | Umami | Fatty |
|---|---|---|---|---|---|---|---|
| PROMPT 1 | Ada | 17.5 | 8.5 | 12.2 | 16.4 | 22.5 | 10.7 |
| | Babbage | 19.5 | 51.1 | 20.2 | 22.0 | 22.6 | 16.0 |
| | Curie | 36.0 | 46.3 | 32.8 | 23.2 | 22.6 | 31.7 |
| | Davinci | 55.0 | 63.2 | 33.3 | 27.2 | **57.0** | 52.0 |
| PROMPT 2 | Ada | 23.1 | 11.9 | 8.5 | 16.8 | -6.4 | 9.9 |
| | Babbage | 27.9 | 55.7 | 19.3 | 23.9 | 29.7 | 34.0 |
| | Curie | 35.4 | 47.9 | 30.3 | 22.7 | 25.4 | 37.5 |
| | Davinci | 50.2 | 54.4 | 34.6 | **28.3** | 49.8 | 42.1 |
| | DeBERTa | **69.1** | **67.0** | **43.9** | 24.7 | 34.4 | **64.0** |
| | Bi-enc$_{MSCG+GKB}$ | 27.4 | -4.4 | 15.4 | 14.4 | -11.8 | 12.6 |
| | Bi-enc$_{ChatGPT}$ | 60.3 | 47.1 | 40.4 | 9.0 | 40.7 | 40.2 |

Table 1: Ranking using conditional probability (Spearman $\rho$%). Prompt1: "[food item] tastes [property]". Prompt 2: "it is known that [food item] tastes [property]".

| Item | Gold | Davinci |
|---|---|---|
| Cracker with Nutella spread | 5 | 324 |
| Chocolate with nut | 15 | 201 |
| Sweet pancake with maple syrup | 34 | 279 |
| Fruit cake | 41 | 265 |
| Sweet cookies with chocolate | 60 | 233 |
| Cracker with jam | 67 | 268 |
| Cooked bell pepper | 252 | 8 |
| Redcurrant | 282 | 36 |
| Radish | 431 | 97 |
| Mascarpone cheese | 477 | 38 |
| Cooked green cabbage | 512 | 49 |
| Saint-agur Cheese | 584 | 61 |

Table 2: Qualitative analysis of the predictions by *davinci* for sweetness, using prompt 1. The table shows the rank positions of several food items, when ranking the items from the sweetest (rank 1) to the least sweet (rank 590), according to the ground truth and the predictions obtained with the Davinci model.

the GPT-3 models, between model size and performance, with the best results achieved by *davinci*. Similar as was observed for the colour domain by Liu et al. (2022a), there seems to be a qualitative change in performance between LLMs such as davinci and smaller models. While there are some differences between the two prompts, similar patterns are observed for both choices. ChatGPT and GPT-4 were not able to provide a ranking of the 590 items, and are thus not considered for this experiment. We also tried ChatGPT on a subset of 50 items, but could not achieve results which were consistently better than random shuffling.

Surprisingly, we find that the DeBERTa model outperforms *davinci* in most cases, and often by a substantial margin. This is despite the fact that

|  |  | Mass $\rho$ | Mass Acc | Height Acc | Size Acc |
|---|---|---|---|---|---|
| COND. PROB. | Ada | 23.0 | 47.8 | 68.7 | 59.1 |
|  | Babbage | 48.9 | 80.9 | 67.9 | 76.4 |
|  | Curie | 30.6 | 65.1 | 77.6 | 86.4 |
|  | Davinci | 36.2 | 76.8 | 76.4 | 80.4 |
| PERPLEXITY | Ada | - | 49.0 | 49.7 | 36.5 |
|  | Babbage | - | 55.0 | 59.1 | 66.7 |
|  | Curie | - | 56.6 | 43.3 | 45.5 |
|  | Davinci | - | 70.8 | 54.7 | 51.1 |
| RANK | ChatGPT[†] | 28.6 | 68.3 | 89.9 | 84.3 |
|  | GPT-4[†] | **58.6** | **84.9** | **99.1** | **99.1** |
|  | DeBERTa | -8.9 | 42.8 | 86.6 | 93.9 |
|  | Bi-enc$_{MSCG+GKB}$ | 31.1 | 69.2 | 69.7 | 71.9 |
|  | Bi-enc$_{ChatGPT}$ | 11.8 | 67.6 | 77.2 | 60.3 |

Table 3: Results for physical properties, viewed as a ranking problem (mass) and as a pairwise judgment problem (mass, height and size). Prompt: "In terms of [mass/height/size], it is known that a typical [concept] is [heavy/tall/large]". Results with [†] required manual post-processing of predictions.

the model is 2 to 3 orders of magnitude smaller. Furthermore, we can see a large performance gap between the two variants of the bi-encoder model, with the model trained on ChatGPT examples outperforming *curie*, and even *davinci* in two cases.

Table 2 presents some examples of the predictions that were made by *davinci* for sweetness (using prompt 1), comparing the ranks according to the ground truth (*gold*) with the ranks according to the *davinci* predictions. The table focuses on some of the most egregious mistakes. As can be seen, *davinci* fails to identify the sweetness of common foods such as chocolate, fruit cake and jam. Conversely, the model significantly overestimates the sweetness of different cheeses and vegetables.

**Physical Properties** Table 3 summarises the results for the physical properties. For mass, we consider both the problem of ranking all objects, evaluated using Spearman $\rho\%$, and the problem of evaluating pairwise judgments, evaluated using accuracy. Height and size can only be evaluated in terms of pairwise judgments. To obtain conditional probabilities from the GPT-3 models, we used a prompt of the form "In terms of [mass/height/size], it is known that a typical [concept] is [heavy/tall/large]". We also tried a few variants, which performed worse. To compute perplexity scores, for evaluating pairwise judgements, we used a prompt of the form "[concept 1] is heavier/taller/larger than [concept 2]". For the baselines, we obtained scores

for the properties *heavy*, *tall* and *large*.

The correlation between model size and performance is far from obvious here, except that *ada* clearly underperforms the three larger models. However, among the GPT-3 models, *babbage* actually achieves the best results in several cases. The results based on conditional probabilities are consistently better than those based on perplexity. ChatGPT and GPT-4 were difficult to use with the ranking prompt, as some items were missing, some were duplicated, and many items were paraphrased in the ranking. The results in Table 3 were obtained after manually correcting these issues. With this caveat in mind, it is nonetheless clear that GPT-4 performs exceptionally well in this experiment. In accordance with our findings in the taste domain, DeBERTa performs very well on the height and size properties, outperforming all GPT-3 models by a clear margin. For mass, however, DeBERTa failed completely, even achieving a negative correlation. The bi-encoder models perform well on height and size, although generally underperforming the largest GPT-3 models. For mass, the bi-encoder trained on ChatGPT examples performs poorly, while the model trained on Microsoft Concept Graph and GenericsKB was more robust. It is notable that the results in Table 3 are considerably higher than those obtained by Li et al. (2023) using OPT (Zhang et al., 2022). For mass, for instance, even the largest OPT model (175B) was not able to do better than random guessing.

In Table 3, the pairwise judgments about mass were assessed by predicting the probability of the word *heavy* (for the GPT-3 models) or by predicting the probability that the property *heavy* was satisfied (for the baselines). Another possibility is to use the word/property *light* instead, or to combine the two probabilities. Let us write $p_{heavy}$ to denote the probability obtained for *heavy* (i.e. the conditional probability of the word, as predicted by the language model, or the probability that the property is satisfied, as predicted by a baseline model), and similar for $p_{light}$. Then we can also predict the relative mass of items based on the value $p_{heavy} \cdot (1 - p_{light})$ or based on the value $p_{heavy}/p_{light}$. These different possibilities are evaluated in Table 4. As can be seen, there is no variant that consistently outperforms the others.

**Analysis of Training Data Overlap** For the baselines, we may wonder to what extent their knowledge comes from the pre-trained language model,

| | ada | babbage | curie | davinci | DeBERTa | Bi-enc$_{\text{MSCG+GKB}}$ | Bi-enc$_{\text{ChatGPT}}$ |
|---|---|---|---|---|---|---|---|
| $p_{heavy}$ | 46.6 | **81.7** | **64.7** | **76.8** | **67.7** | **69.2** | 42.9 |
| $1 - p_{light}$ | 49.8 | 51.3 | 41.5 | 53.4 | 43.4 | 61.9 | **53.4** |
| $p_{heavy} \cdot (1 - p_{light})$ | 46.6 | **81.7** | **64.7** | **76.8** | 47.8 | **69.2** | 48.1 |
| $p_{heavy}/p_{light}$ | **61.4** | 68.3 | 52.1 | 65.9 | 65.2 | 75.7 | 46.5 |

Table 4: Analysis of alternative strategies for predicting pairwise judgements about mass (accuracy).

| | Bitter $\rho$ | Sour $\rho$ | Mass $\rho$ | Height $Acc$ | Size $Acc$ |
|---|---|---|---|---|---|
| Full training | 24.7 | 43.9 | -8.9 | 86.6 | 93.9 |
| Filtered training | 24.8 | 35.0 | 30.7 | 82.0 | 90.8 |

Table 5: Comparison of the DeBERTa model in two settings: the full training setting, where the McRae and CSLB datasets are used for fine-tuning, and a filtered setting, where relevant properties are omitted.

and to what extent it has been injected during the fine-tuning step. For this analysis, we focus in particular on the DeBERTa model, which was fine-tuned on the McRae and CSLB datasets. These datasets indeed cover a number of physical properties, as well as some properties from the taste domain. Table 5 summarises how the performance of the DeBERTa model is affected when removing the most relevant properties from the McRae and CSLB training sets, which we refer to as *filtered training* in the table. For instance, for the property *bitter*, in the filtered setting we omit all training examples involving the properties "bitter" and "can be bitter in taste"; for *sour* we remove the properties "sour" and "can be sour in taste"; for *mass* we remove the properties "heavy", "light", "light weight" and "can be lightweight"; for *height* we remove the properties "short", "can be short", "tall" and "can be tall"; and for *size* we remove the properties "large" and "small". Note that the McRae and CSLB datasets do not cover any properties that are related to sweetness, saltiness, umami and fattiness. The results in Table 5 show that filtering the training data indeed has an effect on results, although the performance of the model overall remains strong. Interestingly, in the case of *mass*, the filtered setting leads to clearly improved results.

## 5 Conclusions

We proposed the use of a dataset from the taste domain for evaluating the ability of LLMs to learn perceptually grounded representations. We found that LLMs can indeed make meaningful predic-

tions about taste, but also showed that a fine-tuned DeBERTa model, and in some cases even a fine-tuned BERT-large bi-encoder, can outperform GPT-3. The performance of these smaller models crucially depends on the quality of the available training data. For this reason, we explored the idea of collecting training data from ChatGPT, using a new prompting strategy. We complemented our experiments in the taste domain with an evaluation of physical properties, where we achieved considerably better results than those reported in the literature (Li et al., 2023). Whereas previous work was essentially aimed at understanding the limitations of language models, our focus was more practical, asking the question: *can high-quality conceptual space representations be distilled from LLMs*? Our experiments suggest that the answer is essentially positive, but that new approaches may be needed to optimally take advantage of the knowledge that can be extracted from such models.

## Limitations

It is difficult to draw definitive conclusions about the extent to which cognitively meaningful representations can be obtained by querying LLMs. Among others, previous work has found that performance may dramatically differ depending on the prompt which is used; see e.g. (Liu et al., 2022a). We have attempted to make reasonable choices when deciding on the considered prompts, through initial experiments with a few variations, but clearly this is not a guarantee that our prompts are close to being optimal. However, this also reinforces the conclusion that LLMs are difficult to use directly for learning conceptual spaces. While we believe that taste represents an interesting and under-explored domain, it remains to be verified to what extent LLMs are able to capture perceptual features in other domains.

**Acknowledgments** This work was supported by EPSRC grant EP/V025961/1.

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

## A Collecting Concept-Property Pairs using ChatGPT

To obtain training data for the bi-encoder model using ChatGPT, we used the following prompt:

> *I am interested in knowing which properties are satisfied by different concepts. I am specifically interested in properties, such as being green, being round, being located in the kitchen or being used for preparing food, rather than in hypernyms. For instance, some examples of what I'm looking for are: **1. Sunflower, daffodil, banana are yellow 2. Guitar, banjo, mandolin are played by strumming or plucking strings 3. Pillow, blanket, comforter are soft and provide comfort 4. Car, scooter, train have wheels 5. Tree, log, paper are made of wood 6. Study, bathroom, kitchen are located in house**. Please provide me with a list of 50 such examples.*

We repeated this request with the same prompt around 10 times. After this, we changed the examples that are given (shown in bold above). This process was repeated until we had 287K concept-property pairs. After removing duplicates, a total of 109K such pairs remained. We found it was necessary to regularly change the examples provided to ensure the examples were sufficiently diverse and to avoid having too many duplicates. These examples were constructed manually, to ensure their accuracy and diversity. Asking for more than 50 examples with one prompt became sub-optimal, as the model tends to focus on a narrow set of similar properties when the list becomes too long.

To verify the quality of the generated dataset, we manually inspected 500 of the generated concept-property pairs. In this sample, we identified 6 errors, which suggests that this dataset is of sufficient quality for our intended purpose. Compared to resources such as ConceptNet, the main limitation of the ChatGPT generated dataset is that it appears to be less diverse, in terms of the concepts and properties which are covered. We leave a detailed analysis of this dataset for future work.

## B Issues with ChatGPT and GPT-4

For the experiments in Table 3, we used ChatGPT and GPT-4 to rank 49, 25 and 26 unique objects according to their mass, height and size respectively. The prompt used was as follows: "Rank the following objects based on their typical [mass/height/size], from [heaviest to the lightest/ tallest to the shortest/ largest to the smallest]", followed by a list of the items. We could not directly evaluate the responses of ChatGPT and GPT-4 because of the following issues:

- Missing objects: For instance, GPT-4 ranked 48 out of 49 objects and ChatGPT ranked 46 out of 49 objects respectively, according to their mass.

- Paraphrasing: While ranking, both GPT-4 and ChatGPT changed some of the the names of the objects. For instance, "wooden train track" was renamed as "wooden train track piece", "gardening shears" was renamed as "garden shears".

- Duplicates: GPT-4 and ChatGPT both occasionally introduced duplicates in the list.

To address these issues, we removed the duplicates and appended the missing items at the end of the ranking. We manually corrected the modified names.