# OpenReview forum: "Cabbage Sweeter than Cake? Analysing the Potential of Large Language Models for Learning Conceptual Spaces"
_EMNLP/2023/Conference — EMNLP 2023 Main_

### Official Review · Reviewer_bmxE · 2023-08-03

**Typos Grammar Style And Presentation Improvements:** Line 034 -> Gärdenfors
**Soundness:** 4

**Excitement:**

4: Strong: This paper deepens the understanding of some phenomenon or lowers the barriers to an existing research direction.

**Paper Topic And Main Contributions:**

This paper seeks to discover whether the taste properties of objects, in addition to their physical qualities, can be be recovered from large language models. The question is interesting because earlier models did not show this ability due to reporting bias in the data. The authors find that GPT-3 and GPT-4 are able to reproduce human ratings on taste and physical properties with reasonable although varying accuracy. For taste properties, the GPTs are not as good as the supervised baselines.



**Reasons To Accept:**

- The paper is a nice demonstration of the capability of LLMs to learn semantic properties of objects.
- The paper provides potentially useful data on which properties are easier or harder to learn (eg bitterness is harder than sweetness).
- Models are prompted in a careful way, using probabilities of completions rather than just a sample, and using a ranking and a pairwise comparison methodology.
- The results are straightforward to interpret.

**Reasons To Reject:**

- The findings are unlikely to move the needle for people who are unconvinced that LLMs can meaningfully learn semantics.
- The authors are not able to determine exactly why the LLMs succeed whereas smaller ones failed.

**Reproducibility:**

4: Could mostly reproduce the results, but there may be some variation because of sample variance or minor variations in their interpretation of the protocol or method.

**Reviewer Confidence:**

4: Quite sure. I tried to check the important points carefully. It's unlikely, though conceivable, that I missed something that should affect my ratings.

---

> ### Author Rebuttal · Authors · 2023-08-25
>
> We thank the reviewer for their careful and supportive review.
>
> Note that our focus is not on showing that LLMs can “understand” the meaning of concepts. Rather, we address a practical question: can we learn meaningful conceptual space representations with the help of LLMs.

---

### Official Review · Reviewer_k2sy · 2023-08-05

**Soundness:** 2

**Excitement:**

2: Mediocre: This paper makes marginal contributions (vs non-contemporaneous work), so I would rather not see it in the conference.

**Paper Topic And Main Contributions:**

The goal of this paper is to determine whether LLMs encode conceptual spaces. Conceptual spaces (a theory associated with Gardenfors) are a type of distributional semantics but one in which the dimensions of the vectors correspond to symbolic perceptual primitives (e.g., hue and brightness for color). This paper asks whether LLMs encode conceptual spaces, looking at a new dataset in the taste domain, as well as some existing datasets on size and mass. The authors use various prompting techniques in order to get LLMs to rank objects along a given dimension (e.g., ranking foods according to sweetness) and then report correlations with human judgments. They show that LLMs have decent correlations (often in the 0.2-0.5 range) and that fine-tuned roberta performs the best.

Overall, I really like the direction of this paper, but have some issues with the experimental design, and, more so, with the theoretical framing w.r.t. conceptual spaces. I elaborate on these concerns below. I'd prefer to see the paper go through a round of revisions before being published.

**Questions For The Authors:**

N/A see weaknesses above

**Reasons To Accept:**

There is a lot of debate and discussion right now around if/how LLMs can encoded information about the nonlinguistic world. Connecting LLMs to theory on conceptual spaces is interesting and a worthwhile direction. The authors also introduce a new dataset on taste which seems interesting and others might like to study.


**Reasons To Reject:**

I have some concerns/confusion around both the experimental design and the theoretical connections the authors want to make.

Theory:
* I am struggling a bit with how/why we should be interpreting the results of the experiments as evidence that LLMs encode conceptual spaces, and even more so with what we should actually require in order for something to count as having a conceptual space. From my understanding of the Gardenfors work, this is a line of distributional semantics theory that is notable because the dimensions of the vectors correspond to sensory (symbolic) primatives. It is often (as I have cited it and seen it cited) contrasted directly with distributional semantics models from NLP which are derived from text, because by-definition the dimensions of the space for text-based DSMs reflect word co-occurances and not perceptual primitives. So, this is to say: in some ways, no matter what, LLMs just by-definition are not conceptual spaces. What makes a conceptual space a conceptual space is that it is not derived from text. That said, I think your project could still be enlightening if you could show that, e.g., the space that the LLMs learn is _isomorphic_ to the conceptual space, similar to the Abdou et al paper. This seems like it would be a very interesting finding. But from what I can gather from your experiments, we cannot conclude that, since the experiments themselves just measure word co-occurances/LM probabilities directly. Thus, I think from the presented experiments, all we can conclude is that LLMs representations are correlated with perceptual features. This isn't necessarily news, but still could be worth documenting. However, I would like to see the paper rewritten with more nuance around the conceptual spaces theory, especially if published without changes to the experiments.

Experimental design:
* The methods you use to get rankings are arguably not measures of the representations directly (as you claim in the intro). Rather, all your experiments rely on e.g., LM continuations or perplexity scores. These don't tell us much about the representation of a word itself (e.g., the representation of the word "cheese") but rather tell us about how the model uses the representation to perform the task of language modeling. To make claims about the representations themselves, e.g., to try to show an isomorphism between the LLM space and the conceptual space, I think you need to work with the word representations themselves. This is not trivial in a multilayered contextualized LM (the way it used to be easy to do with e.g., word2vec). But I think you could find a way. Then, you might consider using something like relational similarity analysis (RSA) in order to measure whether the LLM space and the conceptual space have similar geometries.
* Roberta performs well when fine-tuned on McRae. Have you looked at whether there is overlap between the facts included in McRae and those included in your test sets? I wouldn't be shocked if McRae contains some facts about size or taste...


**Reproducibility:**

4: Could mostly reproduce the results, but there may be some variation because of sample variance or minor variations in their interpretation of the protocol or method.

**Reviewer Confidence:**

4: Quite sure. I tried to check the important points carefully. It's unlikely, though conceivable, that I missed something that should affect my ratings.

---

> ### Author Rebuttal · Authors · 2023-08-25
>
> We thank the reviewer for their careful review.
>
> The reviewer has misunderstood a fundamental point about our paper. Our aim is *not* to analyse whether the representations learned by LLMs themselves can be viewed as (being isomorphic to) conceptual spaces. We fully agree that for showing the latter, a different experimental methodology would be needed. Rather, we analyse whether LLMs can be used, in some way, to learn conceptual space representations (regardless of how concepts are internally represented by LLMs).
>
> “What makes a conceptual space a conceptual space is that it is not derived from text”: different assumptions are made about conceptual spaces by different authors. If we talk about conceptual spaces exclusively as a theory for describing cognitive phenomena, then perhaps this statement has merit. But conceptual spaces have long been studied, including by Gärdenfors (already in the 1990s), as a practical tool for integrating neural and symbolic representations. From this perspective, conceptual space representations can be learned from data (including text). What separates them from other types of vector representations is their structure, and in particular the use of perceptually meaningful dimensions/features.
>
> “All we can conclude is that LLMs representations are correlated with perceptual features”: This is indeed the central point of the analysis. If we can estimate perceptual features with sufficient accuracy, then we can learn meaningful approximations of conceptual spaces, which could play an important role in areas such as explainable AI.
>
> The point in the introduction about evaluating representations is not about evaluating the internal representations of an LLM (see above), but about evaluating learned conceptual space representations. The idea is that we should go beyond using classification tasks, and instead try to evaluate quality dimensions more directly.
>
> To verify the impact of overlapping properties, we have now carried out a few additional experiments, where relevant properties are removed from the McRae and CSLB datasets when training the DeBERTa model. Overall, we can see some impact on the performance, but our main conclusions remain unaffected (and in some cases the results of the DeBERTa model actually improve):
>
> * For the height experiment, we now removed the following properties "short", "can be short", "tall" and "can be tall" from McRae/CSLB. After removing these properties, the performance of the DeBERTa model drops from 86.6 to 82.0
> * For the size experiment, we now removed the properties "large" and "small", after which the performance dropped from 93.9 to 90.8
> * For the mass experiment, we now removed the properties "heavy", "light", "light weight" and "can be lightweight". This, in fact, led to an increase in performance from -8.9 to 30.7 (Spearman rho).
> * For the taste experiments, the properties "bitter" and "can be bitter in taste" were removed from McRae/CSLB, leading to a slight increase from 24.7 to 24.8 for the bitterness property. When removing the "sour" and "can be sour in taste" from McRae/CSLB, we observed a drop from 43.9 to 35.0 for sourness. No overlapping properties occur in McRae/CSLB for sweetness, saltiness, fattiness and umami.
>
> We will add this analysis, to clarify the impact of overlapping properties in the pre-training datasets. However, we would argue that the presence of relevant properties is not in itself problematic. McRae and CSLB are general purpose pre-training datasets, which were selected independent of the considered evaluation tasks. Even if the pre-training datasets contain some relevant properties, we would expect LLMs to outperform this DeBERTa baseline. The fact that this is not always the case highlights the limitations of directly prompting LLMs for learning conceptual spaces.

---

### Official Review · Reviewer_BTQu · 2023-08-10

**Soundness:** 3

**Excitement:**

3: Ambivalent: It has merits (e.g., it reports state-of-the-art results, the idea is nice), but there are key weaknesses (e.g., it describes incremental work), and it can significantly benefit from another round of revision. However, I won't object to accepting it if my co-reviewers champion it.

**Paper Topic And Main Contributions:**

This paper probes the question: can high-quality conceptual space representations be distilled from LLMs?
The authors examine an unexplored space of taste and add the physical properties space. They present a way to utilize GPT to artificially generate data. They find that DeBERTa is at least comparable to the much larger GPT-based models.

**Questions For The Authors:**

* The paragraph of "Ranking with ChatGPT and GPT-4" (starting at line 159): Did you run GPT several times to verify the responses are consistent?
* The paragraph of "Baseline: Bi-encoder" (starting at line 185): Did you verify the generated data is satisfactory?
* Why do you think that DeBERTa is better than GPT? Maybe some more comprehensive error analysis would be better here.
* Table 2: I have no idea what these numbers mean. Why are they not normalized?

**Reasons To Accept:**

* This paper presents an interesting exploration of yet unexplored conceptual spaces. I like this human-inspired analysis direction.
* The authors focus on an interesting commonsense problem that suffers deeply from reporting bias, and show where it is possible to trust LLMs.
* The paper is mostly well-written.
* The choice of models is satisfactory.

**Reasons To Reject:**

* Table 1: Some error analysis due to the surprising results would be much better than just stating them and acknowledging they are indeed surprising.
* As I did not fully understand Table 2, I cannot judge it.
* The conclusion "The key to achieving good results with such smaller models is to have access to suitable training data." is rather trivial. I wonder what is the real contribution of this paper besides being an interesting analysis.

**Reproducibility:**

4: Could mostly reproduce the results, but there may be some variation because of sample variance or minor variations in their interpretation of the protocol or method.

**Reviewer Confidence:**

3: Pretty sure, but there's a chance I missed something. Although I have a good feel for this area in general, I did not carefully check the paper's details, e.g., the math, experimental design, or novelty.

**Typos Grammar Style And Presentation Improvements:**

* Lines 178-179: I would add an example to help the reader understand the data better.
* Table 1: Mark in bold the best-performing model per column.

---

> ### Author Rebuttal · Authors · 2023-08-25
>
> We thank the reviewer for their careful review.
>
> Some error analysis about the results from Table 1 is presented in Table 2. This table shows the rank positions of 590 different food items, according to the ground truth and according to the Davinci predictions. For instance, the sweetest food item would receive a rank of 1. The examples illustrate how Davinci makes substantial and surprising errors, e.g. predicting chocolate or sweet cakes to be among the least sweet food items, while predicting vegetables such as cabbage to be among the most sweet food items. We will clarify how the table has to be interpreted, and add an expanded error analysis to the supplementary materials.
>
> The point in the conclusions section about high-quality training data is not that it is required, but rather that it is sufficient, i.e. that smaller models can be surprisingly strong when the right kind of training data is available. We will rephrase the sentence to make this clearer.
>
> Note that it is quite common for conferences such as EMNLP to publish analysis papers. We thus believe that it is unfair to criticise the paper for not having “any real contribution apart from an interesting analysis”.
>
> The full set of ChatGPT and GPT-4 results was only generated once, because we had to manually edit the responses, as explained in Appendix C. In initial experiments we found the results to be very stable, although there are of course small differences across different runs. However, these small differences would not affect our main conclusions, which are that GPT-4 and ChatGPT are difficult to use, but seem to perform at a level that is considerably above the other models when it is possible to use them.
>
> Regarding the training set that was generated using ChatGPT, please note that we have made the entire dataset available in the supplementary materials. We found this dataset to be highly accurate upon inspection: in a sample of 500 examples from the dataset, we only found 6 errors.
>
> The main limitation of the GPT-based models is that there is no obvious way to directly exploit their latent representations. This means that we have to resort to proxies, such as evaluating the perplexity of artificially generated statements, which appears to be sub-optimal. Our hypothesis is therefore that such models are best used for generating training examples (which is much better aligned with their training objective), while BERT-based models may still be better for learning representations.

---

### Meta-Review · Area_Chair_QfuG · 2023-09-19

**Recommendation:** 3

**Metareview:**

The paper probes to what extent LMs encode conceptual spaces in the sense of Peter Gardenfors’ theory, using a new dataset on the taste domain. The works is interesting and focuses on the problem of how competence of LMs can be related to interpretable semantic dimensions, such as the ones in the Conceptual Space theory. On other hand, one reviewer raises important methodological issues I agree with. These concern the experiments and especially their interpretation, and  MUST be carefully addressed in the revised version of the paper.

---

### Decision · Program_Chairs · 2023-10-07

**Decision:**

Accept-Main

**Comment:**

The paper probes to what extent LMs encode conceptual spaces in the sense of Peter Gardenfors’ theory, using a new dataset on the taste domain. The works is interesting and focuses on the problem of how competence of LMs can be related to interpretable semantic dimensions, such as the ones in the Conceptual Space theory. On other hand, one reviewer raises important methodological issues I agree with. These concern the experiments and especially their interpretation, and  MUST be carefully addressed in the revised version of the paper.